Infrared thermography applied to lower limb muscles in elite soccer players with functional ankle equinus and non-equinus condition

Rodríguez-Sanz David davidrodriguezsanz@gmail.com 1
Losa-Iglesias Marta Elena marta.losa@urjc.es 2
López-López Daniel daniellopez@udc.gal 3
Calvo-Lobo César cecalvo19@hotmail.com 4
Palomo-López Patricia patibiom@unex.es 5
Becerro-de-Bengoa-Vallejo Ricardo ribebeva@ucm.es 6
1 Physical Therapy & Health Sciences Research Group; Physiotherapy Department, Faculty of Health, Exercise and Sport, European University , Madrid , Spain
2 Faculty of Health Sciences, Universidad Rey Juan Carlos , Madrid , Spain
3 Research, Health and Podiatry Unit, Department of Health Sciences, Faculty of Nursing and Podiatry, Universidade da Coruña, Ferrol, Spain
4 Nursing and Physical Therapy Department, Institute of Biomedicine (IBIOMED), Universidad de León , León , Spain
5 University Center of Plasencia, Universidad de Extremadura, Plasencia , Spain
6 School of Nursing, Physiotherapy and Podiatry, Universidad Complutense de Madrid, Madrid, Spain
Gill Jason
Electronic publication date: 2017 May 25
Publication date: 2017
Volume: 5
Electronic Location ID: e3388
Received 2017 Mar 7; Accepted 2017 May 8
Copyright: ©2017 Rodríguez-Sanz et al.
Copyright year: 2017
Copyright holder: Rodríguez-Sanz et al.
License: This is an open access article distributed under the terms of the Creative Commons Attribution License, which permits unrestricted use, distribution, reproduction and adaptation in any medium and for any purpose provided that it is properly attributed. For attribution, the original author(s), title, publication source (PeerJ) and either DOI or URL of the article must be cited.
License URL: https://creativecommons.org/licenses/by/4.0/

Keywords: Sport, Thermography, Ankle, Foot

Funding: The authors received no funding for this work.

==============================
Gastrocnemius-soleus equinus (GSE) is a foot-ankle complaint in which the extensibility of the gastrocnemius (G) and soleus muscles (triceps surae) and ankle are limited to a dorsiflexion beyond a neutral ankle position. The asymmetric forces of leg muscles and the associated asymmetric loading forces might promote major activation of the triceps surae, tibialis anterior, transverses abdominal and multifidus muscles. Here, we made infrared recordings of 21 sportsmen (elite professional soccer players) before activity and after 30 min of running. These recordings were used to assess temperature modifications on the gastrocnemius, tibialis anterior, and Achilles tendon in GSE and non-GSE participants. We identified significant temperature modifications among GSE and non-GSE participants for the tibialis anterior muscle (mean, minimum, and maximum temperature values). The cutaneous temperature increased as a direct consequence of muscle activity in GSE participants. IR imaging capture was reliable to muscle pattern activation for lower limb. Based on our findings, we propose that non-invasive IR evaluation is suitable for clinical evaluation of the status of these muscles.

Introduction

Gastrocnemius-soleus equinus (GSE) is a functional limitation of the ankle. GSE is defined as the inability of the ankle to dorsiflex beyond a neutral position with the knee extended (it remains <0°) or with the knee flexed (it remains <0°) (after excluding osseus restriction) (DiGiovanni et al., 2002; Downey & Banks, 1989). Although GSE is clinically asymptomatic, it may cause an alteration in the triceps surae muscles, tibialis anterior muscle (TA), and Achilles tendon (AT). GSE is closely related to asymmetric loading patterns and muscular alteration for contraction (Lamm, Paley & Herzenberg, 2005; Silfverskiöld, 1924; Downey, 1992; Root, Orien & Weed, 1977).

Asymmetric body loading is associated with GSE and may cause a higher activation of low-back and pelvic muscles (e.g., the quadratus lumborum) and lower limb muscles (e.g., the TA). GSE has been much studied using the mean of force pressure platform, Romberg-test analysis, and sway-area development approaches (Blustein & D’Amico, 1985; Mahar, Kirby & McLeod, 1985; Bhave, Paley & Herzemberg, 1999; Blake & Ferguson, 1992). The relationships between muscle contraction condition and posture have been investigated by electromyography (Vink & Huson, 1987; Gurney et al., 2001; Balestra et al., 2001). GSE condition shows a relationship with gait and posture (Bhave, Paley & Herzemberg, 1999; Blake & Ferguson, 1992). Also, several studies have investigated whether activation of skeletal muscles promotes heat transfer and thus increases cutaneous temperature (Merla et al., 2005; Zontak et al., 1998; Kurt & Damiano, 2016; Formenti et al., 2017).

Orthostatic posture is typically assessed by the complex activation of anti-gravitational muscles (e.g., the triceps surae muscles). Modifications in cutaneous temperature are associated with gait and posture alterations, asymmetric loading forces, and an altered range of ankle or knee movement. High thermal infrared (IR) imaging can non-invasively identify cutaneous temperature variations (Merla et al., 2005).

Here we aimed to assess whether high-resolution thermal IR can detect cutaneous temperature variations in GSE and Non-GSE individuals and thus detect association between the extensibility to triceps-surae (with GSE and Non-GSE) and the role of TA in professional sportsmen. The aim of the study was to check thermal differences between GSE and Non GSE-group before running and after running exercise.

Figure 1 Anterior view thermacam of tibialis anterior muscle.

Methods

In this case-control study, 21 healthy male participants (professional soccer players) (age 23 ± 2.9 years, body mass index (applying Quetelet’s equation follow BMI = weight (kg) / height(m)2) 20.2 ± 1.2) were included. A consecutive sampling method was used to select participants. Eighteen subjects were right-footed, and three were left-footed, as defined by the self-report on the lower limb predominant used in sports activities. All participants successfully completed the study. The exclusion criteria were the presence of musculoskeletal injuries, low back and pelvic pain, sprains, tendon injuries in lower limbs, use of drugs in the week preceding the test, and scoliosis.

The principal study variable was temperature. An IR ThermaCam was used to register the maximum, minimum and mean temperature values of the TA, AT, and G regions.

Ethical considerations

The Research and Ethics Committee of the University of A Coruña (A Coruña, Spain; record number: CE 06/2014) approved the study. All subjects provided informed consent before the beginning of the study. The ethical standards for human experimentation of the Declaration of Helsinki were respected. The Strengthening the Reporting of Observational Studies in Epidemiology (STROBE) guidelines were applied. The study size was derived as a convenience sample method.

IR imaging

The subjects were given a series of standardization rules before attending the experimental measurements (Merla et al., 2005). Specifically, during the week prior to the trial, the participants were asked not to use drugs. Also, on the test day, vasomotor substances (e.g., caffeine) and heavy meals were not allowed.

Paper signals were attached to the lower limb with anatomical references (TA, G, and AT). All measurements were acquired in a laboratory with a temperature of 24.1 ± 1 C°, humidity of 45% ± 10% and no direct ventilation-flow towards participants or raters.

Clinical exploration and capture of IR-imaging

First, participants lay in a supine position on a gurney, and their GSE status was assessed with the knee extended and flexed. The range of ankle dorsiflexion movement was checked using a goniometer to evaluate the angle between the plantar line of the foot and the tibia axis. The rater maintained the subtalar joint in a neutral position and the angle in a dorsiflexion position (rather than midfoot-dorsiflexion (rocker bottom) or midfoot-equinus (pseudoequinus)). The Silverskiold-test was used to check GSE from other types of equines (Lamm, Paley & Herzenberg, 2005; Silfverskiöld, 1924). A 20° movement with the knee flexed and 10°  movement with the knee extended was considered a normal range of ankle dorsiflexion (Lamm, Paley & Herzenberg, 2005; Downey, 1992; Root, Orien & Weed, 1977). Equinus analysis was carried out by the same Podiatry Doctor (R.B.B.V) to ensure the reliability of measurements.

The IR-imaging process (high resolutions thermograms) started with the participant standing up in a relaxed position. We captured the TA, G, and AT regions in that order. Five IR-images were taken for every muscle. Participants then ran for 30 min on a treadmill at a speed of 9 km/h and the IR-imaging repeated.

IR-imaging was performed using an FLIR/SC3000/QWIP Thermacan infrared thermal device with a 8–9 µm spectral range and 0.02-K temperature sensitivity (NETD at 30 °C). The 320 × 240/FPA device presents a 20°-lens. The images were captured with a 1.1 mrad spatial-resolution. IR-imaging acquisition was carried out by the same clinician (DRS) using a tripod.

Data Analysis

IR-images and data were analyzed using rater (DRS) with the FLIR® software Termocan Researcher Professional V.2.9 (FLIR, Wilsonville, OR, USA). This software provides to rater the minimum, maximum and mean thermal values for the selected muscles. This body selections-regions were bilaterally selected with landmark position and selected by the blinded clinical operator who ignored hypothesis of study and the experimental conditions. An IR-Imaging example is shown in Fig. 1.

Table 1 Participant characteristics (n = 10 non-equinus and 11 equinus participants).

	Equinus group	Non- equinus group	
Age (years)*	23.56 ± 2.61 (21–28)	20 ± 2.16 (19–27)	
Height (cm)*	175.5 ± 7.9	173.1 ± 5.2	
Weight (kg)*	75.2 ± 3.9	74.5 ± 3.1	
Body mass index*	20.2 ± 1.3	20.1 ± 1.5	
Notes.

* No statistically significant difference between groups (p⩾0.05).

Table 2 Temperature values (Degrees Celsius) for tibialis anterior, gastrocnemius-soleus and Achilles tendon for gastrosoleus equinus (GSE) and non-gastrosoleus equinus (Non-GSE) participants before exercise.

Variable	Mean	SD	P value	
Tibialis right anterior minimum temperature before exercise	
GSE	31.01	±1.3	.342*	
Non GSE	30.86	±0.56	.331*	
Tibialis anterior right maximum temperature before exercise	
GSE	30.85	±1.32	.397*	
Non GSE	31.07	±0.81	.401*	
Tibialis anterior right mean temperature before exercise	
GSE	31.3	±1.3	.430*	
Non GSE	31.09	±0.56	.435*	
Tibialis anterior left minimum temperature before exercise	
GSE	28.82	±2.47	1.14*	
Non GSE	29.74	±0.82	1.15*	
Tibialis anterior left maximum temperature before exercise	
GSE	31.87	±1.35	.486*	
Non GSE	31.65	±0.53	.504*	
Tibialis anterior left mean temperature before exercise	
GSE	31.01	±1.3	.354*	
Non GSE	31.86	±0.56	.366*	
Gastrocnemius left minimum temperature before exercise	
GSE	28.41	±1.57	.449*	
Non GSE	28.71	±1.39	.441*	
Gastrocnemius left maximum temperature before exercise	
GSE	30.85	±1.32	.442*	
Non GSE	31.07	±0.81	.452*	
Gastrocnemius left mean temperature before exercise	
GSE	29.92	±1.34	.687*	
Non GSE	30.12	±0.84	.681*	
Gastrocnemius right minimum temperature before exercise	
GSE	28.35	±2.0	1.065*	
Non GSE	29.1	±0.94	1.1*	
Gastrocnemius right maximum temperature before exercise	
GSE	31.1	±1.14	.357*	
Non GSE	31.91	±0.86	.362*	
Gastrocnemius right mean temperature before exercise	
GSE	29.89	±1.14	.608*	
Non GSE	30.17	±0.93	.614*	
Achilles tendon left minimum temperature before exercise	
GSE	22.52	±3.59	.581*	
Non GSE	23.22	±1.16	.605*	
Achilles tendon left maximum temperature before exercise	
GSE	29.52	±1.58	.819*	
Non GSE	28.93	±1.75	.815*	
Achilles tendon left mean temperature before exercise	
GSE	26.57	±1.92	.393*	
Non GSE	26.87	±1.47	.399*	
Achilles tendon right minimum temperature before exercise	
GSE	22.74	±2.34	.364*	
Non GSE	23.05	±1.26	.367*	
Achilles tendon right maximum temperature before exercise	
GSE	29.57	±1.88	.062*	
Non GSE	29.51	±1.13	.064*	
Achilles tendon right mean temperature before exercise	
GSE	26.29	±1.68	1.48*	
Non GSE	27.46	±1.93	1.47*	
Notes.

* No statistically significant difference between groups (p⩾0.05).

N = 10 non-equinus and N = 11 equinus participants.

Statistical Analysis

Statistical analyses were performed with SPSS (version 22.0 for Windows, IBM SPSS Statistics for Windows; IBM, Armonk, NY, USA) with an α error of 0.05 (95% confidence interval (CI)), with the desired power of 80% (β error of 0.2).

A Shapiro–Wilks test was used to assess data normality. All data were normally distributed, and parametric statistical tests were selected. The mean and standard deviation of the temperature data were obtained for the selected lower limb muscles (AT, TA,G).

Unpaired sample student’s t test were performed to test for statistically significant differences in height, weight, body mass index, and age between the two groups. A Paired Student’s t-tests were performed to determine differences between the groups (equinus vs. non-equinus), as well as between imaging sessions (before vs. after running).

Results

We found no statistically significant differences between the equinus vs. non-equinus groups for participant height, weight, age or body mass index (Table 1).

We found no significant differences in TA, G or AT temperatures between the GSE and non-GSE participants before running (Table 2).

However, after running, the TA temperature (minimum, maximum, and mean) was significantly warmer in the GSE than the Non GSE participants (P < .05). Also, the minimum G temperatures (both right and left) and left AT mean temperature were significantly warmer in the GSE than the Non-GSE participants (Table 3).

Table 3 Temperature values (Degrees Celsius) for tibialis anterior, gastrocnemius-soleus and Achilles tendon for gastrosoleus equinus (GSE) and non-gastrosoleus equinus (non-GSE) participants after exercise.

Variable	Mean	SD	P value	
Tibialis anterior right minimum temperature after exercise	
Non GSE	27.1	±1.9	.007†	
GSE	29.04	±0.76	.008†	
Tibialis anterior right maximum temperature after exercise	
Non GSE	30.28	±1.53	.006†	
GSE	31.95	±0,8	.006†	
Tibialis anterior right mean temperature after exercise	
Non GSE	29.14	±1.88	.035†	
GSE	30.63	±0.89	.034†	
Tibialis anterior left minimum temperature after exercise	
Non GSE	27.12	±1.63	.001†	
GSE	29.42	±0.65	.001†	
Tibialis anterior left maximum temperature after exercise	
Non GSE	30.52	±1.19	.038†	
GSE	31.72	±1.25	.038†	
Tibialis anterior left mean temperature after exercise	
Non GSE	29.36	±1.62	.029†	
GSE	30.77	±1.04	.027†	
Gastrocnemius left minimum temperature after exercise	
Non GSE	27.92	±1.65	.033†	
GSE	29.38	±1.16	.031†	
Gastrocnemius left maximum temperature after exercise	
Non GSE	30.31	±1.39	.036†	
GSE	31.32	±1.22	.035†	
Gastrocnemius left mean temperature after exercise	
Non GSE	29.27	±1.53	.063*	
GSE	30.44	±1.12	.06*	
Gastrocnemius right minimum temperature after exercise	
Non GSE	27.81	±1.67	.025†	
GSE	29.29	±0.97	.024†	
Gastrocnemius right maximum temperature after exercise	
Non GSE	30.5	±1.32	.140*	
GSE	31.3	±0,98	.135*	
Gastrocnemius right mean temperature after exercise	
Non GSE	29.20	±1.48	.025†	
GSE	30.55	±0.93	.023†	
Achilles tendon left minimum temperature after exercise	
Non GSE	24.04	±3.34	.146*	
GSE	25.75	±1.24	.139*	
Achilles tendon left maximum temperature after exercise	
Non GSE	30.44	±2.02	.278*	
GSE	31.23	±0.93	.267*	
Achilles tendon left mean temperature after exercise	
Non GSE	27.88	±2.13	.020†	
GSE	29.87	±1.3	.019†	
Achilles tendon right minimum temperature after exercise	
Non GSE	24.41	±3.15	.216*	
GSE	25.85	±1.65	.207*	
Achilles tendon right maximum temperature after exercise	
Non GSE	30.27	±1.79	.073*	
GSE	31.5	±1.02	.07*	
Achilles tendon right mean temperature after exercise	
Non GSE	27.91	±1.67	.006*	
GSE	29.85	±1.2	.005*	
Notes.

* No statistically significant difference between groups (p⩾0.05).

† Statistically significant difference between groups (P < 0.05). N = 10 non-equinus and N = 11 equinus participants.

Discussion

Here, we identified an increase in TA temperature after running in professional soccer professionals with GSE condition compared to those without GSE. The minimum G temperatures (right and left lower limb) and mean AT (left lower limb) temperature were also higher in GSE than non-GSE participants.

Ankle torque was higher in GSE participants than non-GSE participants (Wrobel, Connolly & Beach, 2004). The TA requires stronger contraction in GSE condition and, therefore, might be affected early by fatigue, thus explaining our observed increase in TA temperature in GSE participants compared to non-GSE participants. While participants remain without running activity, we didn’t found differences. The running exercise may serve a stimulus to increase temperature in muscles and consequently differences between groups.

Researchers have addressed the necessary degrees of ankle dorsiflexion and basal values (DiGiovanni et al., 2002; Root, Orien & Weed, 1977). Biomechanically, the maximum ankle dorsiflexion during the stance-phase of a normal gait occurs before heel lift with the knee extended (DiGiovanni et al., 2002). The minimum ankle range of motion for normal gait is 10° dorsiflexion and 20° plantarflexion (DiGiovanni et al., 2002; Downey & Banks, 1989; Root, Orien & Weed, 1977). The most deeply known range of movement values for ankle dorsiflexion in the reviewed literature for static evaluation that the minimum dorsiflexion movement for the ankle for normal gait is 10° of motion (Lamm, Paley & Herzenberg, 2005; Mcglamry & Kitting, 1973; Knutzen & Price, 1994; Nuber, 1988; Lavery, Armstrong & Boulton, 2002; Wrobel, Connolly & Beach, 2004; Winter, 1984). GSE produces a higher loading force to the foot and can lead to foot-ankle biomechanical pathologic (e.g., plantar fasciitis, pes planus, hallux abductus valgus, Achilles tendinosis, Charcot’s midfoot collapse, and diabetic ulcerations) (Lamm, Paley & Herzenberg, 2005). DiGiovanni et al. (2002) found GSE in patients with foot and ankle pain. However, GSE is also found in asymptomatic patients (Brodersen, Pedersen & Reimers, 1993).

Further studies will be needed to improve our knowledge of muscle condition and to establish the clinical relevance of the association between temperature and cutaneous muscle projection (Abate et al., 2010). Based on our findings, we propose that IR-imaging can be a reliable tool for clinical therapeutic assessment.

Conclusions

GSE participants had a higher TA muscle temperature after exercise that non-GSE participants. The GSE groups showed a higher TA skin temperature. Therefore infrared thermography, in the way it measures skin temperature, could serve as screening tool for preventing or therapeutic actions. Further research is needed to identify other factors associated with GSE condition, as well as to better understand the factors that contribute to different temperature pattern in the lower limbs.

Supplemental Information

Supplemental Information 1 Strobe

Click here for additional data file.

Supplemental Information 2 Raw data

Click here for additional data file.

Additional Information and Declarations

Competing Interests

Author Contributions

Human Ethics

Data Availability

The authors declare there are no competing interests.

David Rodríguez-Sanz conceived and designed the experiments, performed the experiments, wrote the paper, reviewed drafts of the paper.

Marta Elena Losa-Iglesias conceived and designed the experiments, prepared figures and/or tables.

Daniel López-López analyzed the data, contributed reagents/materials/analysis tools, wrote the paper.

César Calvo-Lobo analyzed the data, wrote the paper.

Patricia Palomo-López contributed reagents/materials/analysis tools, prepared figures and/or tables.

Ricardo Becerro-de-Bengoa-Vallejo conceived and designed the experiments, performed the experiments, prepared figures and/or tables.

The following information was supplied relating to ethical approvals (i.e., approving body and any reference numbers):

The Research and Ethics Committee of University of A Coruña (A Coruña, Spain; record number: CE 06/2014) approved the study.

The following information was supplied regarding data availability:

The raw data has been supplied as a Supplementary File.

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
