# Peer review of "Infrared thermography applied to lower limb muscles in elite soccer players with functional ankle equinus and non-equinus condition"

_PeerJ, doi:10.7717/peerj.3388_

## Round 0.1 · original submission · Major Revisions

Two reviewers have carefully reviewed your paper and found merit in this work. They have, however a number of concerns about the paper in its present form and have suggested that you make a number of major revisions to paper. Please could you ensure that you address all of their comments in your revised version. In particular, please ensure that you present the data in the excel spreadsheet more clearly and check that the temperatures are reported correctly.

·

Basic reporting

Reasonably simple experimental design, well executed.
A few minimal "English" errors
Raw data is difficult to analyse in it's current form, presumably copied from a number of spreadsheets.

Experimental design

Good use of template to ensure accurate measurement positions, and use of single rater.

Validity of the findings

As it is presented the findings do not appear to be correct.
I believe the data has been inverted GSE vs non-GSE

Additional comments

Obviously of clinical significance, would have benefitted from more review before submission.
I'm sure your observations are correct but the data does not confirm this.

Reviewer 2 ·

Basic reporting

The study is sufficiently readable, but in some cases the used terms are not specific for the context. Nevertheless, language can be improved.

The introduction section contains all the key elements of the study. However, it is very concise, and the key points (such as the relationship between posture alterations and cutaneous temperature modifications) result only cited and not well connected between each other. I suggest to dedicate more space to explain deeply the relationship between GSE, posture, cutaneous temperature, thermography.
Cited literature within the whole manuscript is pertinent, but I suggest referring to some more papers regarding methodological aspects related to the analysis of thermal images and the skin temperature distribution:
Kurt Ammer, Damiano Formenti. Editorial: Does the type of skin temperature distribution matter?. Thermology International. 26(2):51-54. 2016.
Damiano Formenti, Nicola Ludwig, Alessio Rossi, Athos Trecroci, Giampietro Alberti, Marco Gargano, Arcangelo Merla, Kurt Ammer, Andrea Caumo. Skin temperature evaluation by infrared thermography: Comparison of two image analysis methods during the nonsteady state induced by physical exercise. Infrared Physics and Technology. 81:32-40. 2017.

In the manuscript there is one figure (Figure 1) and 3 tables. Figure 1 seems out of focus. Is it possible to substitute it with another image?
Graphic appearance of the tables should be improved. CI column can be deleted. What does the term “Bilat” mean). I suppose it should be deleted. The real obtained p-values are expressed, so substitute “<” with “=”. Delete also the column N, and I suggest to insert the n values of both groups in the legend.

Experimental design

The aim of this study was to investigate the skin temperature modifications of triceps surae and tibialis anterior in response to a 30 min running exercise in subjects with gastrocnemius equinus condition compared to a control group. The topic of the study is really interesting. However, the authors do not state the research question clearly, as the hypothesis is not present. Also, it is not clear how this study can contribute to fill a gap in the literature, if the gap is present. In the introduction the authors should state what this study add to the existing literature. The final part of the introduction should express clearly the aim of the study, as well the hypothesis.

Methodology is not accurately described. I have some important concerns that should be addressed:
Line 110: insert unit of measurement of body mass index.
Line 117: why did the authors considered maximum, minimum and mean temperature of the region of interests? Which is the advantage of using these three temperature parameters? Authors present the three parameters in the tables and results, however they do not consider to discuss such results, and whether there are differences between the three temperature parameters.
Line 128: please delete “To improve the accuracy of the imaging”.
Line 140: “ensure” instead of “improve”.
Line 143-144: why did the authors choose 30 min on a treadmill at 9 km/h? Why not another exercise? Why not 20 min at 8 km/h? Please justify the choice of this protocol.
Line 150: “…were analysed by the same rater (DRS) using the FLIR…”
This is the major point that I have to rise: Statistical analysis and data presentation.
Line 169: Statistical analysis seems inappropriate. In order to test the difference between the two groups, an unpaired sample student’s t test should be performed. A paired student’s t test should be performed to test the same group before and after running.
However, I would suggest to perform a two way mixed anova (group x time) with repeated measurements on one factor (time), where group is GSE and nonGSE, and time is before and after running. This test should be performed for right and left limb separately, thus having 2 two way mixed anova.
Line 174: Results. Results section is very poorly presented. I suggest to describe with more details the main findings together with data of the tables.

Validity of the findings

Certainly, the present findings are interesting and the conclusion described by the authors are coherent to the results presented. However, since statistical analysis seems inappropriate, I would be more cautious in drawing certain conclusion.
For example, in Line 214, supposing to have the same results with the appropriate statistical analysis, I would propose: “GSE groups showed a higher TA skin temperature. Therefore, Infrared thermography, in the way it measures skin temperature, could serve as screening tool for preventing or therapeutic actions.”
Line 217: please delete “abnormal” and “distribution”. What is considered abnormal and what normal? Moreover, since the authors did not evaluate the variability of the pixels temperature within a region of interest, it may be misleading. In fact, it usually refers to the distribution of the pixel temperatures within a region of interest.
Moreover: It is not discussed why no differences between the two groups were found before running, and why significant differences were found after running. Did the running exercise serve a stimulus? Which are the underlying physiological bases? I would suggest to provide further possible explanation of these findings within the discussion.

External reviews were received for this submission. These reviews were used by the Editor when they made their decision, and can be downloaded below.

---

## Round 0.2 · accepted · Accept

The reviewers feel that you have addressed their concerns, so I am happy to accept your paper now.

Reviewer 2 ·

Basic reporting

Please pay attention to the references format, in particular to the references 16 and 17.
Appearance of the tables (e.g. lines) can be improved.

Experimental design

In Statistical analysis, line 140 should be: "Unpaired Student’s t-tests were performed to determine differences between the groups (equinus vs. non-equinus), and paired Student's t-tests were used to investigate differences between imaging sessions (before vs. after running).

Please insert unit of measurement of body mass index in table 1.

Validity of the findings

No comment

Additional comments

The authors provided sufficient responses to all the reviewer comments.

External reviews were received for this submission. These reviews were used by the Editor when they made their decision, and can be downloaded below.